

# CNV-P: a machine-learning framework for predicting high confident copy number variations

Taifu Wang[1,*], Jinghua Sun[1,2,*], Xiuqing Zhang[1,2,3], Wen-Jing Wang[1] and Qing Zhou[1]

[1] BGI-Shenzhen, Shenzhen, China
[2] College of Life Sciences, University of Chinese Academy of Sciences, Beijing, China
[3] Guangdong Enterprise Key Laboratory of Human Disease Genomics, Beishan Industrial Zone, Shenzhen, China
* These authors contributed equally to this work.

## ABSTRACT

**Background:** Copy-number variants (CNVs) have been recognized as one of the major causes of genetic disorders. Reliable detection of CNVs from genome sequencing data has been a strong demand for disease research. However, current software for detecting CNVs has high false-positive rates, which needs further improvement.

**Methods:** Here, we proposed a novel and post-processing approach for CNVs prediction (CNV-P), a machine-learning framework that could efficiently remove false-positive fragments from results of CNVs detecting tools. A series of CNVs signals such as read depth (RD), split reads (SR) and read pair (RP) around the putative CNV fragments were defined as features to train a classifier.

**Results:** The prediction results on several real biological datasets showed that our models could accurately classify the CNVs at over 90% precision rate and 85% recall rate, which greatly improves the performance of state-of-the-art algorithms. Furthermore, our results indicate that CNV-P is robust to different sizes of CNVs and the platforms of sequencing.

**Conclusions:** Our framework for classifying high-confident CNVs could improve both basic research and clinical diagnosis of genetic diseases.

## INTRODUCTION

Copy number variations (CNVs) are one of the genetic variations and sequence polymorphisms that widely exist in the human genome. Research shows that CNVs are closely related to the pathogenesis and development of many human diseases such as autism, Parkinson's and other neurological diseases (*Hollox et al., 2008*; *Pankratz et al., 2011*; *Rosenfeld et al., 2010*; *Sebat et al., 2007*). Therefore, accurate detection of CNVs is essential for the diagnosis and research of such diseases.

With the rapid development of high-throughput sequencing technology, genomic sequencing-based technology for CNVs detection has gradually become a leading method

Corresponding authors
Wen-Jing Wang,
wangwenjing@genomics.cn
Qing Zhou, zhouqing1@genomics.cn

owing to its high speed, high resolution, and high repeatability. Many sequencing-based CNVs detection methods have been proposed (*Kosugi et al., 2019*; *Pirooznia, Goes & Zandi, 2015*; *Zhao et al., 2013*). Typical CNVs detection approaches mainly utilize three signatures to detect CNVs: read depth (RD), read pairs (RP), split read (SR) (*Pirooznia, Goes & Zandi, 2015*). RD means the number of reads that encompass or overlap CNVs. For example, a deletion indicates a decrease in the average depth of this area. RP refers to the distribution of the insert size of the sequenced library. If the mapping distance of read pairs significantly deviates from the average value of the sequencing library, such discordant alignment features herald the occurrence of CNVs. SR indicates the split (soft-clipped) alignment features of reads that span CNVs. The initial strategies for detecting CNVs mainly focused on one of these features (*Abyzov et al., 2011*; *Chen et al., 2009*; *Medvedev et al., 2010*). Most of the approaches have high false-positive rates because of the noises of sequencing data, such as sequencing error and artificial chimeric reads. Ambiguous mapping of reads from repeat- or duplication-rich regions also decreases the accuracy of CNVs (*Kosugi et al., 2019*; *Teo et al., 2012*). Consequently, tools integrating multiple features to detect CNVs have been gradually developed (*Bartenhagen & Dugas, 2016*; *Layer et al., 2014*; *Rausch et al., 2012*), while their performance still needs be further modified (*Kosugi et al., 2019*).

To identify high confident CNVs, a commonly used strategy is setting a cutoff value or applying various statistical distributions to filter fragments. This strategy greatly depends on the expertise of researchers and their subjective assumptions about the analyzed data. Another strategy uses the intersection of CNVs generated by two or more algorithms. However, due to various CNV-property-dependent and library-property-dependent features used by different detection methods, they usually provide inconsonant results. Thus, a large number of potentially true CNVs could be discarded. Additionally, some tools, such as MetaSV (*Mohiyuddin et al., 2015*), Parliament2 (*Zarate et al., 2020*) and FusorSV (*Becker et al., 2018*), use the method of integrating and merging CNVs from multiple software. These approaches require output results of several certain tools, usually more than four software, while reanalyzing CNVs using their default methods is impractical and time-consuming. Recently, some software using machine learning model to detect or predict CNVs have been developed, such as GATK-SV (*Werling et al., 2018*) and CNV-JACG (*Zhuang et al., 2020*), it is still necessary to develop more accurate tools.

Here, we developed a machine-learning framework for CNVs prediction (CNV-P), aiming to accurately predict CNVs from the results of present software. CNV-P collected three aforementioned signatures (RD, RP and SR) and other information of the putative CNVs. The results of our model on real data demonstrate that CNV-P greatly improves the performance of state-of-the-art algorithms.

## MATERIALS & METHODS

### Data download and preprocessing

The gold-standard sets of CNVs from nine individuals (NA19238, NA19239, NA19240, HG00512, HG00513, HG00514, HG00731, HG00732, HG00733) were downloaded from

*Chaisson et al. (2019)*. The whole genome sequencing (WGS) data (~30×) of these nine individuals were downloaded from the National Center for Biotechnology Information (NCBI) with an accession number SRP159517 (Tables S1, S2). For external validation samples, the sequencing data of NA12878 and HG002 were also downloaded from NCBI with accession numbers SRP159517 and SRP047086 respectively. The gold-standard CNVs of NA12878 were generated by three data sets: the Database of Genomic Variants (http://dgv.tcag.ca/dgv/app/home?ref=GRCh37/hg19) (*Macdonald et al., 2013*), the 1000 Genomes Project Phase III (https://www.internationalgenome.org/phase-3-structural-variant-dataset) (*Sudmant et al., 2015*), and the CNVs of PacBio data from *Pendleton et al. (2015)*. The gold-standard CNVs of HG002 were downloaded from *Zook et al. (2019)*.

For the above gold-standard sets, we excluded other types of CNVs except for deletion and duplication, removed CNVs shorter than 100 base pair (bp) and merged fragments with over 80% reciprocal overlaps. On average, each sample had more than 10,000 CNVs after processing (Table S1). For WGS data, the clean reads after removing adapter and filtering low-quality reads were aligned to the human genome reference (hg19) with bwa (*Li, 2013*) 'mem' command to generate the BAM file. All of these datasets were generated by standard WGS protocol, with libraries of approximate 400 bp insert size and average ~30× coverage (Table S2).

## Generate simulated dataset

We generate random CNVs (range from 100 bp to 100 kilobase (kb)) based on a copy of human genome (hg19) using mason2 (*Holtgrewe, 2010*). To avoid the same or similar CNVs between training data and test data, we selected fragments on chromosome 1 and chromosome 2 as training samples and CNVs on chromosome 3 and chromosome 4 as testing samples (more details in Table S1). Then, the paired-end sequencing reads (100 bp) from the altered genome was simulated by wgsim (*Li, 2011*), with an insert size of 500 bp and 0.001 base error rate.

## Training set and test set

We chose five common software to obtain the initial sets of CNVs for simulated data and the downloaded sequencing data (deletions and duplications): Lumpy (*Layer et al., 2014*), Manta (*Chen et al., 2015*), Pindel (*Ye et al., 2009*), Delly (*Rausch et al., 2012*) and breakdancer (*Chen et al., 2009*). The details of running parameters were shown in the supplemental methods section. The original CNVs were then performed as follows:
1. Removed other types of CNVs except for deletion and duplication. 2. Removed CNVs with >10 bp overlapped with N region of human genome (download from http://genome.ucsc.edu/). 3. Merged CNVs with ≥80% reciprocal overlaps and kept the union part of fragments. 4. Removed CNVs that less than 100bp. Then, we labeled these treated CNVs as either 'True' or 'False' based on their overlapped part with gold-standard CNVs. CNVs having ≥80% reciprocal overlap with the gold-standard CNVs in simulated data were labeled as "True" and the cutoff was set to ≥50% for sequencing data. We then selected data of six individuals as a training set and the other three samples as a test set, including two dependent validation datasets (more details in Tables S1, S2).
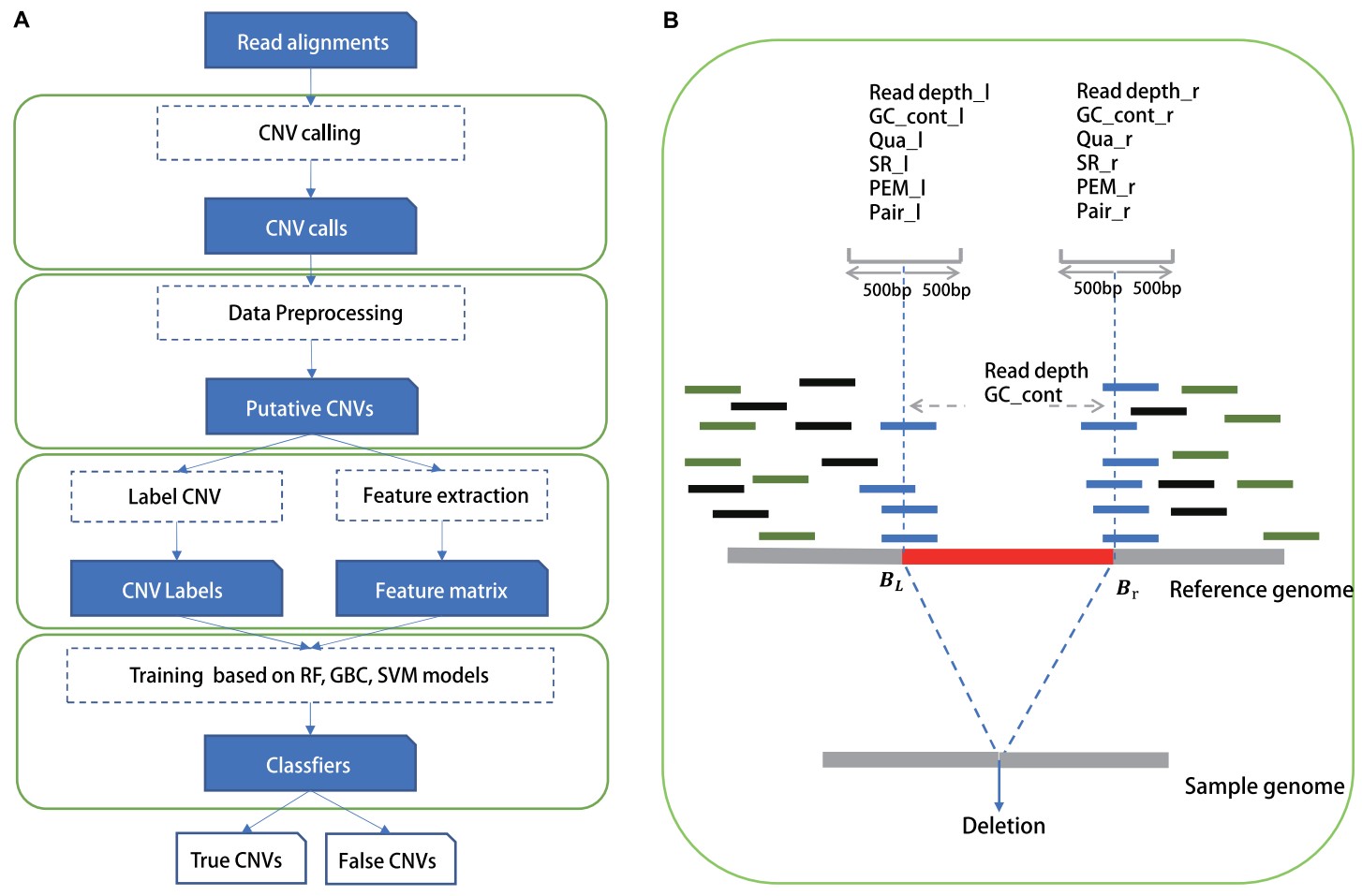

**Figure 1** **The study overview of CNV-P.** (A) The workflow of CNV-P framework classifyng candidate CNVs as True or False. (B) The features we used to train supervised machine learning models.

## Feature extraction

We chose commonly used signals by detection tools as features in our training model, such as read depth, information of paired and spited read, mapping quality and GC content of CNVs, as well as all these features around CNV's boundaries (Table S3). Training features were obtained from the alignment results (Fig. 1) in BAM format, which was generated by a read aligner that supports partial read alignments, such as BWA-MEM (*Li, 2013*). For read depth-based and GC content-based features, we computed the read depth and GC rate of three regions: 500 bp upstream and downstream of the left breakpoint $L_{b1k}$, 500 bp upstream and downstream of the right breakpoint $R_{b1k}$, and the region from start to end $C_{start-end}$. Read depth was calculated by total number of aligned bases divided by the length of the region. We then normalized the read depth by the average coverage of entire genome and processed log2 transformation to eliminate the impact of fluctuations in sequencing depth. GC content was also calculated in these three regions. Thus, using read depth and GC content of the three local regions ($L_{b1k}$, $R_{b1k}$ and $C_{start-end}$), six features were defined. Split-read, read pair and mapping quality were computed for two regions: $L_{b1k}$ and $R_{b1k}$. Split read-based features were defined as the

number of clipped reads within the area $L_{b1k}$ or $R_{b1k}$. Read pair–based features were defined as the number of outlier reads pair within $L_{b1k}$ or $R_{b1k}$. Normally, The insert size of a normal paired-end read should be within $m_{is} \pm n\sigma_{is}$, where $m_{is}$ and $\sigma_{is}$ are the median and standard deviation of insert size, respectively, and $n$ is the number of standard deviation from the median (we set is to 3). In addition to aberrant insert size, we also calculated the number of reads without pair within the area $L_{b1k}$ or $R_{b1k}$. The features of mapping quality were defined as the number of reads with mapping quality <10 within $L_{b1k}$ or $R_{b1k}$. Finally, we also normalized the value of split reads, read pair and mapping quality according to the mean value of genome coverage. Besides, we included the size and type of CNVs as training features, since the efficacy of CNVs could vary for different size ranges and types (duplication/deletion).

## Comparison with CNV-JACG, MetaSV and hard cutoff method

We compared the performance of CNV-P with that of CNV- JACG (*Zhuang et al., 2020*), MetaSV (*Mohiyuddin et al., 2015*) and hard cutoff method in the same datasets. Since MetaSV currently does not support Delly's output, only four CNV detection tools (Lumpy, Manta, Pindel, and breakdancer) were taken into consideration. CNV-JACG was conducted running with default parameters (details in supplementary methods). MetaSV was carried out with complete mode. For hard cutoff method, we used SR and RP as the evidence to support the existence of CNVs, therefore, the number of SR and RP greater than 2, 5, and 10 were set as hard cutoff to evaluate. SURVIVOR (*Jeffares et al., 2017*) was used to merge fragments with 80% overlap after filtering by CNV-P, CNV- JACG, MetaSV and hard cutoff method.

## Methodology evaluation

We calculated the classifier performance on the test dataset in terms of precision and recall (TP: true positive, TN: true negative, FP: false positive, FN: False negative)

$$\text{Precision} = \frac{\text{TP}}{\text{TP} + \text{FP}}$$

$$\text{Recall} = \frac{\text{TP}}{\text{TP} + \text{FN}}$$

$$\text{F1 score} = \frac{2 * \text{Precision} * \text{Recall}}{\text{Precision} + \text{Recall}}$$

Also, we plotted the Receiver Operating Characteristic (ROC) curves with the area under curve (AUC) for model evaluation. ROC curves were drawn based on a series of false positive rates (FPR) and true positive rates (TPR).

# RESULTS

## Study overview

In this study, we built a random forest (RF) framework for the CNVs prediction based on both simulated and real datasets (Fig. 1A). Firstly, we identified CNVs using five common tools (Lumpy, Manta, Pindel, Delly and breakdancer). For each set, we removed CNVs

with low quality or locating on the N region of the human genome (details in "Methods"). Secondly, we labeled CNVs as either "True" or "False" based on a 50% reciprocal overlap with the gold-standard CNVs in real data and 80% reciprocal overlap in simulated data respectively (details in "Methods"). Next, we extracted the signatures around these CNVs such as RD, SR and RP as training features from alignment results (Fig. 1B).

We then split the data set into a training set and a test set. Based on the training set, we trained a RF classifier to identify CNVs as "true" or "false". We performed 10-repeated 10-fold cross-validation for optimal parameter selection and used the ROC curve to quantify the prediction performance. Next, we evaluated the robustness of our models on test data from multiple aspects, such as sizes of CNVs and platforms of raw sequencing data. We also compared the performance of Support Vector Machine (SVM) and Gradient Boosting Classifier (GBC) with our random forest model. Finally, we validated our model on two extra data sets.

## Performance of CNV-P on a simulated dataset

We trained RF, GBC, and SVM classifiers for CNV prediction based on a simulated dataset (details in "Methods"). The results showed that there was a significant improvement after CNV-P prediction compared with the original CNV results. The precision of CNVs produced by each CNV detection tools improved from 49.58% to 99%, almost without loss of recall rate (Figs. 2A, 2B). Compared with GBC and SVM classifiers, RF was slightly superior. The RF classifiers for five tools achieved comparable performance since an average increasing of F1-score was about 14.55% for Lumpy, 14.34% for Manta, 13.84% for Pindel, 11.10% for Delly and 16.16% for breakdancer (Fig. 2C).

## Performance of CNV-P on a real dataset

In this part, we trained RF classifier for the five selected tools respectively based on real samples. The 10-repeated 10-fold cross-validation was performed for optimal parameter selection (Fig. S1). The overall diagnostic ability of each classifier was measured as the area under the ROC curve for the test dataset. The highest value of AUC was 97.10% for the model of Lumpy while the model for Pindel had the smallest value of 93.62% (Fig. 3A). Each classifier accurately classified the CNVs as either true or false at 91.76–95.17% precision and 87.75–96.54% recall rate (Fig. 3B). After processing by CNV-P, a large number of false-positive CNVs were removed, and the majority of true CNVs were remained (Fig. 3C).

To dissect the principle of the CNV-P classifier, we assessed the relative importance of each feature for corresponding classifiers. As expected, for all classifiers, read-depth provided the most discriminatory power to make accurate predictions (Fig. S2). However, the second important feature was inconsistent between different classifiers. It was probably due to various detection algorism these tools used.

To evaluate the robustness of CNV-P, we trained each model on various proportions of training data (from 10% to 90% in increments of 20%). The results showed a steady improvement in accuracy (precision and recall rate) with an increase in the number of training data (Fig. S3).

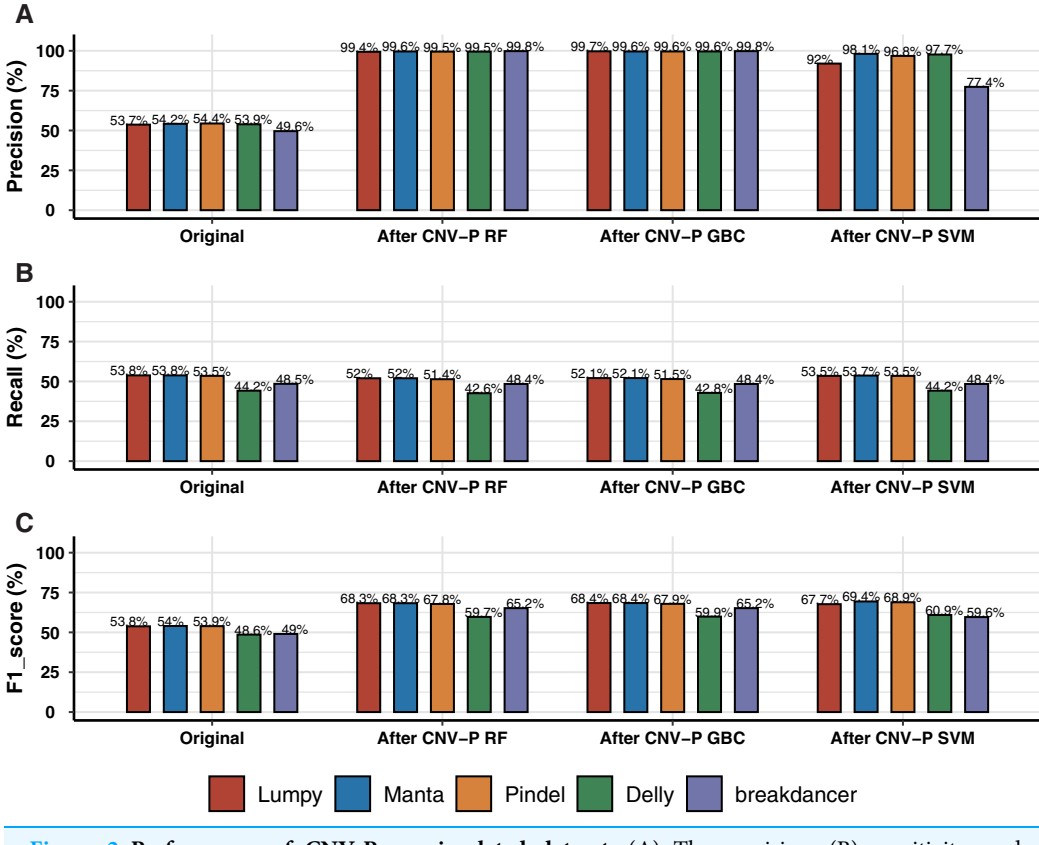

**Figure 2 Performance of CNV-P on simulated dataset.** (A) The precision, (B) sensitivity, and (C) F1-score of CNV-P on simulated dataset.

We further assessed the performance of CNV-P for CNVs of different sizes. We divided CNVs into three sets based on their length: CNV_S (100 bp to 1 kb), CNV_M (1 kb to 100 kb) and CNV_L (>100 kb). The overall precisions were greatly improved, comparing with the raw CNVs achieved by the corresponding software (Fig. 3D). We noticed that almost all precision and recall rates of CNV_S and CNV_M were over 90%, while theses values of CNV_L were slightly lower. These results are probably caused by the insufficient number of CNV_L in our training data.

We also profiled the distribution of predicted probability scores for all CNVs within a different size range. Since CNVs with a probability score >0.5 were classified as true in our CNV-P prediction results, we found that the threshold of 0.5 distinguished true and false CNVs very well (Fig. S4). Besides, the probability scores could be used as a measurement of confidence for a certain fragment of CNVs, which would provide support evidence in further analysis.

Furthermore, we implanted two additional models, GBC and SVM, to train CNV-P classifiers. Comparing the precision and recall values, as well as the result of ROC curve, we found they had comparable performance (Fig. S5). Still, the RF classifier was recommended as the first choice with a slight superiority.

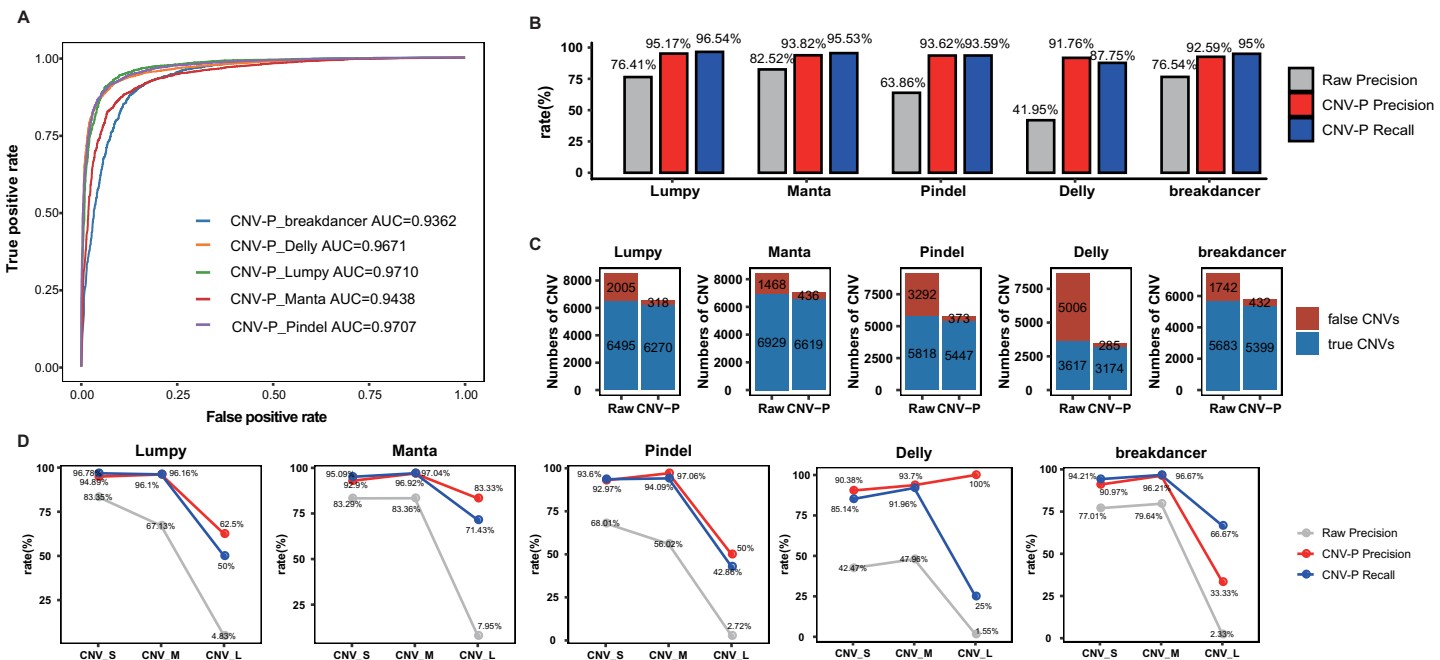

**Figure 3 Performance of CNV-P on real dataset.** (A) Receiver operating characteristic (ROC) curves of CNV-P for five detection tools. (B) The precision and recall rate of CNV-P. (C) The number of CNVs before and after CNV-P predicting for five commonly used tools. (D) The precision and recall rate of CNV-P at different size range of CNVs. CNV_S: 100 bp to 1 kb, CNV_M: 1 kb to 100 kb, CNV_L: >100 kb.

## Prediction on external data sets

To further evaluate the performance of CNV-P, we implemented our models on two independent WGS datasets of NA12878 and HG002 (Table S1). Since we had proved that increasing the size of training data could improve the accuracy of our model (Fig. S3), the final classifiers were trained on both the training set and test set mentioned above. Consistent with the above results, CNV-P produced the optimal performance with AUCs of 0.89–0.95 in NA12878 (Fig. 4A). Most of the false-positive CNVs were removed with a loss of a small number of true positive fragments (Figs. 4B, 4C). Likewise, our approach had a similar performance on sample HG002 (Figs. 4D–4F).

We next compared CNV-P with other post-process tools for CNV filtering, including CNV-JACG and MetaSV. We also included commonly used hard filtering method, setting cut-off of SR and RP number for each CNV. We applied various filtering approaches on NA12878 and HG002, and then evaluated fragments using gold-standard CNVs of these two samples. Our results showed that CNV-P had the highest F1-score among all the post-process methods (Table 1).

Besides, we evaluated the performance of our approach in data generated from multiple sequencing platforms. With precision of 91.6–96.8% and recall rates of 84.1–94% (Fig. 5), CNV-P showed similar performance on sequencing data generated by BGISEQ-500. Moreover, in addition to the trained classifiers for the above five software, we provided extra modules in our approach for training and predicting if CNVs were detected

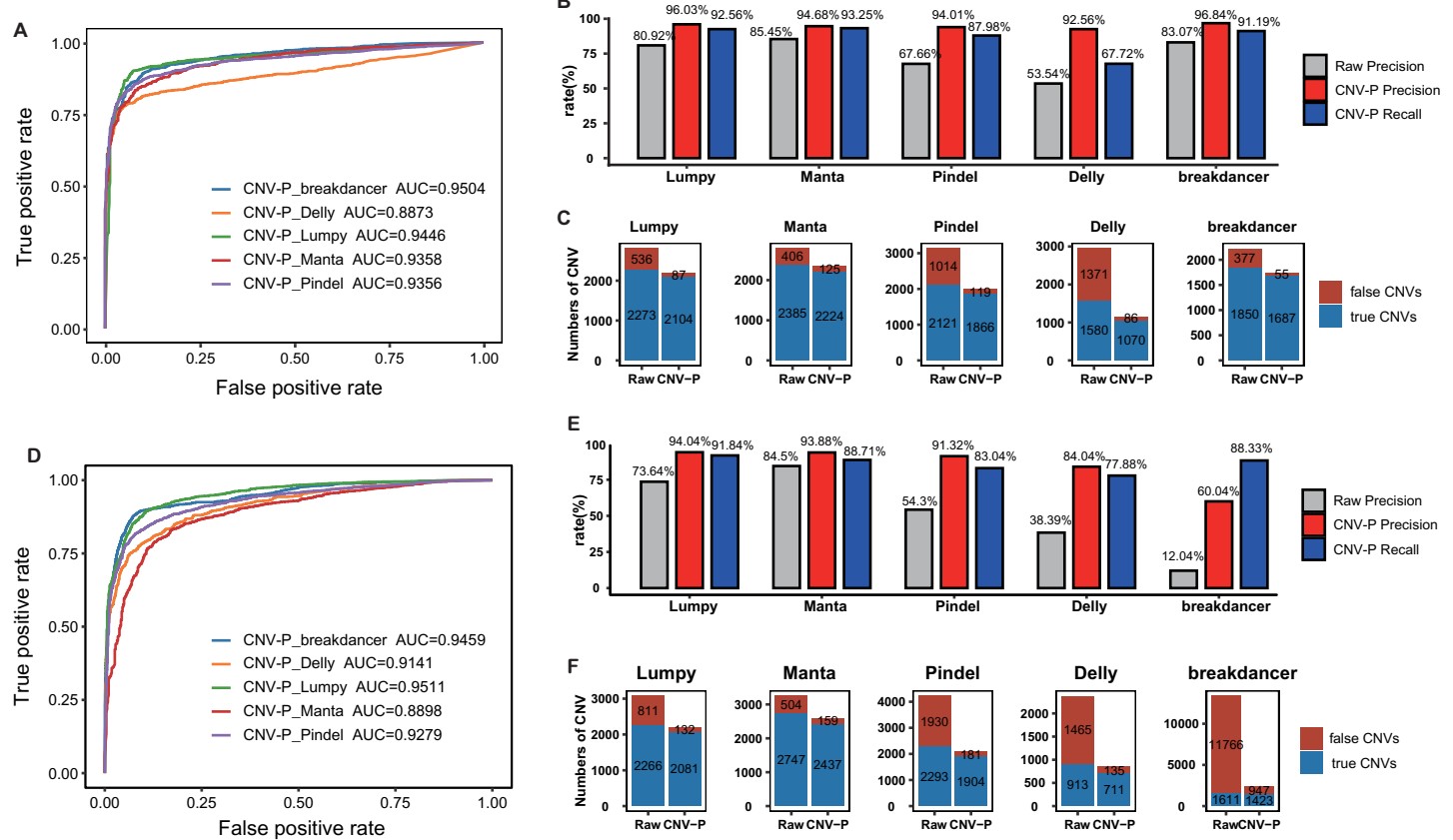

**Figure 4 Performance of CNV-P on other validation dataset.** CNV-P detects high-confident CNVs with high precision and recall rate on two independent sequencing datasets from NA12878 (A, B, C) and HG002 (D, E, F). (A, D) Receiver operating characteristic (ROC) curves of CNV-P. (B, E) The precision and recall rate of CNV-P; (C, F) The number of classified CNVs by CNV-P from five commonly used tools.

**Table 1 Comparison with CNV-JACG, MetaSV and hard cutoff method in NA12878 and HG002.**

| Sample | Method | Precision | Recall | F1-score |
|---|---|---|---|---|
| NA12878 | RAW | 0.6032 | 1.0000 | 0.7525 |
| | Hard_Cutoff_2 | 0.6197 | 0.9792 | 0.7590 |
| | Hard_Cutoff_5 | 0.7145 | 0.8630 | 0.7818 |
| | Hard_Cutoff_10 | 0.7780 | 0.6976 | 0.7356 |
| | CNV-JACG | 0.6828 | 0.7496 | 0.7146 |
| | MetaSV | 0.7094 | 0.8817 | 0.7862 |
| | CNV-P | 0.9007 | 0.7977 | 0.8461 |
| HG002 | RAW | 0.2054 | 1.0000 | 0.3408 |
| | Hard_Cutoff_2 | 0.4026 | 0.9729 | 0.5695 |
| | Hard_Cutoff_5 | 0.5740 | 0.8653 | 0.6901 |
| | Hard_Cutoff_10 | 0.6642 | 0.7482 | 0.7037 |
| | CNV-JACG | 0.5443 | 0.7076 | 0.6153 |
| | MetaSV | 0.5917 | 0.8274 | 0.6900 |
| | CNV-P | 0.7078 | 0.7516 | 0.7290 |

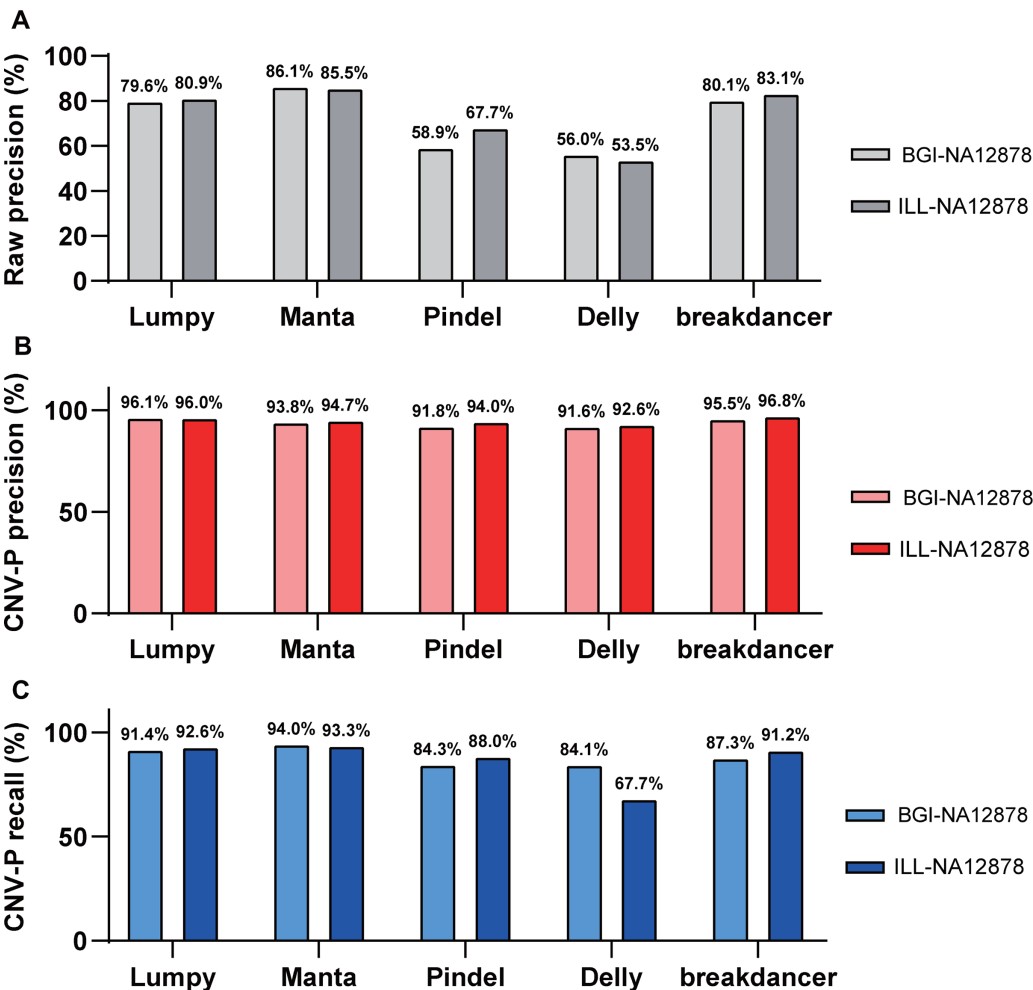

**Figure 5 Performance of CNV-P on different sequencing platform.** The precision and recall rate of CNV-P for sample NA12878 using sequencing data generated from BGISEQ-500 and Illumina. (A) The raw precision results; (B) Precision rate; (C) Recall rate.

by other tools. These results suggest that our approach is suitable for CNVs generated from multiple sequencing platforms and detecting software.

## DISCUSSION

Detecting CNVs from WGS is error-prone because of short-length reads and library-property-dependent bias (*Kosugi et al., 2019*). Inflated false positive makes it a big challenge for researchers to identify clinically relevant CNVs, as it is time- and money-consuming to validate a large amount of false positive CNVs. To solve this problem, we develop CNV-P, an effective machine-learning-based framework to acquire high-confident CNVs. Instead of handling the shortcomings of existing methods by developing another detecting algorithm, CNV-P focuses on providing a reliable set of CNVs from existing detection software. We demonstrate that CNV-P can identify a set of high-confidence CNVs with high precision and recall rates. Moreover, CNV-P is robust to

the proportion of variants in training sets, size of CNVs and sequencing platforms, indicating the utility of CNV-P in a variety of clinical or research contexts.

Comparing with the conventional method of using hard cutoff, such as a minimum number of supporting reads, to filter CNV results, CNV-P greatly reduces errors caused by lack of expertise and subjective assumptions. Instead of running default multiple software in advance, CNV-P can make accurate predictions for each tool dependently. In addition to the five commonly used software that we have trained prediction models, we provide an extra module in CNV-P including the function of model training and predicting if CNVs are detected by other tools.

However, our models may have weaker power for large-size CNVs, because there are only a small number of large fragments in our training data. Besides of data from healthy individuals, we believe that great improvement could be made to identify large-size true CNVs in the future when more datasets are accumulated.

## CONCLUSIONS

CNV-P is a well-performed machine-learning framework for accurately filtering CNVs. CNV-P framework can be applied on CNVs from various detection methods and sequencing platforms, making our framework easy to adopt and customize. CNV-P greatly helps to generate a set of high-confident CNVs, benefiting both basic research and clinical diagnosis of genetic diseases.

## ACKNOWLEDGEMENTS

The authors thank Dr. Jian Guo for constructive comments on this project and Chen Ye for data download and management.

### Funding

This project was supported by the National Key Research and Development Program of China (No.2018YFC1004900), the National Natural Science Foundation of China (No.81300075), the Science, Technology and Innovation Commission of Shenzhen Municipality under grant (No.JCYJ20170412152854656, JCYJ20180703093402288). There was no additional external funding received for this study. The funders had no role in study design, data collection and analysis, decision to publish, or preparation of the manuscript.

### Grant Disclosures

The following grant information was disclosed by the authors:
National Key Research and Development Program of China: 2018YFC1004900.
National Natural Science Foundation of China: 81300075.
Science, Technology and Innovation Commission of Shenzhen Municipality: JCYJ20170412152854656, JCYJ20180703093402288.

## Competing Interests

The authors declare that they have no competing interests.

## Author Contributions

- Taifu Wang conceived and designed the experiments, performed the experiments, analyzed the data, prepared figures and/or tables, authored or reviewed drafts of the paper, and approved the final draft.
- Jinghua Sun performed the experiments, analyzed the data, prepared figures and/or tables, authored or reviewed drafts of the paper, and approved the final draft.
- Xiuqing Zhang conceived and designed the experiments, performed the experiments, authored or reviewed drafts of the paper, and approved the final draft.
- Wen-Jing Wang conceived and designed the experiments, authored or reviewed drafts of the paper, and approved the final draft.
- Qing Zhou conceived and designed the experiments, analyzed the data, prepared figures and/or tables, authored or reviewed drafts of the paper, and approved the final draft.

## Data Availability

The data is available in the Supplemental Materials and CNV-P is available at GitHub: https://github.com/wonderful1/CNV-P.

## Supplemental Information

Supplemental information for this article can be found online at http://dx.doi.org/10.7717/peerj.12564#supplemental-information.

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
