# Peer review of "CNV-P: a machine-learning framework for predicting high confident copy number variations"

_PeerJ, doi:10.7717/peerj.12564_

## Round 0.1 · original submission · Major Revisions

Your manuscript has been reviewed by two experts in the field. As shown in their comments below, both of them basically admit the value of this work while raising several important points; particularly, both request more detailed description in the main text. Please read their comments carefully and revise the manuscript accordingly. Looking forward to your revised manuscript.

Reviewer 1 ·

Basic reporting

In this manuscript, Wang et al. proposed a machine learning framework, CNV-P, for predicting high confident copy number variations (CNV). The computational detection of CNVs using genome sequencing data is an important task and should be of general interest to the community. I think this work may be helpful to the CNV data end-users.

Experimental design

The current CNV calling methods are suffering from low accuracy with high false positive rates. Different from the conventional methods which use hard cutoff to select high confidence callings, the authors proposed a machine learning method based on random forest to predict high confidence CNVs using multiple features such as read depth, split reads and read pair around the putative CNV fragments.

Validity of the findings

The authors validated their method on synthesized and real data sets. The proposed method showed high precisions with good sensitivity.

Additional comments

Minor Comment:
It is interesting to know the contributions by each feature in the random forest prediction model, and compared performance with the hard cutoff method using the eminent feature.

Reviewer 2 ·

Basic reporting

The detailed method of training in real sample is understandable by Table S1 and S2 but it should be written in the method section in the main text with similar details.

Experimental design

no comment

Validity of the findings

Integrate SV calls from multiple algorithms and extract better calls by using supporting evidences based on random forest learning of PE, SR, RD and so on, is not a new idea (for example, Werling et al. Nat Genet 2018; 50: 727-36. or Zhuang et al. NAR genom bioinform 2020; 2: Iqaa071, possibly more?) While the details were somehow different each other.

This reviewer asks the authors to compare the performance. It would also be valuable to see the comparison of the performance with other integrative CV callers such as MetaSV.

---

## Round 0.2 · Minor Revisions

The revised manuscript has been reviewed by the two original reviewers. As you can see from their comments below, one of them is satisfied with your revision while the other points out that a few concerns remain, Please read the comments carefully and re-revise the manuscript accordingly unless you think that the comments are inappropriate.

Reviewer 1 ·

Basic reporting

All my questions are addressed.

Experimental design

All my questions are addressed.

Validity of the findings

All my questions are addressed.

Reviewer 2 ·

Basic reporting

The authors appropriately addressed my concern

Experimental design

no comment

Validity of the findings

The authors mostly addressed my concern, this reviewer appreciated it. However, in my original comment, I also asked the authors to compare with Werling et al. method

(now merged to) https://github.com/broadinstitute/gatk-sv#gather-sample-evidence

Please mention it and compare, or if there is difficulty to compare with it, please explain.

Also this reviewer would request the authors to describe the comparison pipeline at the authors' github space or somewhere, for the readers to be able to replicate it.

---

## Round 0.3 · accepted · Accept

Since the remaining reviewer considers that the revision has been done (to some extent), I would like to recommend its acceptance to the journal. Congratulations!

Reviewer 2 ·

Basic reporting

NA

Experimental design

NA

Validity of the findings

I've checked the authors' github space

https://github.com/wonderful1/CNV-P

and confirmed that they successfully addressed the point of concern.

Additional comments

NA